# Multitask Siamese Network for Remote Photoplethysmography and Respiration Estimation

**DOI:** 10.3390/s22145101

**Published:** 2022-07-07

**Authors:** Heejin Lee, Junghwan Lee, Yujin Kwon, Jiyoon Kwon, Sungmin Park, Ryanghee Sohn, Cheolsoo Park

**Affiliations:** 1Department of Computer Engineering, Kwangwoon University, Seoul 01897, Korea; bobae1998@kw.ac.kr (H.L.); asdfg20564@kw.ac.kr (Y.K.); bbkjo@kw.ac.kr (J.K.); 2Department of Information Convergence, Kwangwoon University, Seoul 01897, Korea; hjn040281@gmail.com; 3Department of Electrical Engineering, Pohang University of Science and Technology, Seoul 37673, Korea; sungminpark@postech.ac.kr; 4Emma Healthcare, Seongnam-si 13503, Korea

**Keywords:** heart rate, respiration rate, contactless technique, deep learning, Siamese network, multitasking

## Abstract

Heart and respiration rates represent important vital signs for the assessment of a person’s health condition. To estimate these vital signs accurately, we propose a multitask Siamese network model (MTS) that combines the advantages of the Siamese network and the multitask learning architecture. The MTS model was trained by the images of the cheek including nose and mouth and forehead areas while sharing the same parameters between the Siamese networks, in order to extract the features about the heart and respiratory information. The proposed model was constructed with a small number of parameters and was able to yield a high vital-sign-prediction accuracy, comparable to that obtained from the single-task learning model; furthermore, the proposed model outperformed the conventional multitask learning model. As a result, we can simultaneously predict the heart and respiratory signals with the MTS model, while the number of parameters was reduced by 16 times with the mean average errors of heart and respiration rates being 2.84 and 4.21. Owing to its light weight, it would be advantageous to implement the vital-sign-monitoring model in an edge device such as a mobile phone or small-sized portable devices.

## 1. Introduction

In 2020, the coronavirus disease 2019 (COVID-19) pandemic introduced many changes in our lives. Given its very high infection rate, the number of infected people increases rapidly, and hospital accommodation facilities often become insufficient. Thus, patients are often required to be self-quarantined at home and monitor their conditions. Some patients suffer from a lack of knowledge regarding these self-checks. Therefore, the importance of non-face-to-face healthcare monitoring has also been emphasized [1,2], and heart rate (HR) and respiration rate (RR) are vital signs that could allow for the monitoring of post-COVID-19 infections. In an experiment which involved 2745 subjects with COVID-19, the HR and RR abruptly increased when the disease symptoms began to appear [3]. This study confirmed that HR and RR could be early indicators of the development of COVID-19 [1,4]. Furthermore, HR is used to detect heart disease. According to the World Health Organization (WHO) 2019 report on human deaths caused by diseases worldwide, heart diseases rank first and chronic obstructive pulmonary disease (COPD) ranks third [3]. Heart diseases with high mortality rates are even more dangerous because they rarely have symptoms and are not treated in a timely manner. This strengthens the importance of the constant monitoring of heart signals [5]. Respiration rate could be also used as an indicator of serious illness since an abnormal RR is often found in several diseases such as the hypercarbia, hypoxia and hypoxemia [6]. The body attempts to correct these disorders by increasing the tidal volume and respiratory rate. Monitoring both the HR and RR could be helpful to scoring sleep stages and diagnosing sleep disorders such as sleep apnea [7,8]. In addition, mental stress can be detected through heart and respiratory signals based on deep neural networks [9].

Among the heart-signal-monitoring approaches, photoplethysmography (PPG) is a method used to detect the amount of blood flow using light sources based on the detection of the amount of blood volume through peripheral blood vessels [10]. PPG signals could be utilized to calculate HR by detecting and counting their peaks over a minute, since a PPG signal peak corresponds to the heartbeat. The respiratory rate (RR) is the frequency of breathing, that is, the number of breaths in a minute. Along with the HR, RR is also a critical vital sign to monitor the health condition, in particular the lung condition, which is affected by infectious diseases such as the flu and COVID-19 [10]. Respiratory signals are recorded during the inhalation and exhalation respiratory phases, and the respiratory rate is calculated using the period of the respiratory signal. This respiratory rate can also be estimated from the PPG signal, as reported in previous studies [11,12]. Thus, this study attempts to simultaneously estimate both the HR and RR through predicting the rPPG signal from facial images. This respiratory rate could also be estimated using the PPG signal, as reported in previous studies [11,12]. So, we decided to use both the PPG and respiratory signals.

Most of the sensors used to measure heart activity contact the body on the finger, wrist, or earlobe [13,14,15]. Respiratory signals can be measured from chest movements using chest-strap equipment [16]. However, the existing sensor systems require additional (costly) devices and are inconvenient to carry in daily life if continuous health monitoring is necessary. Therefore, this study suggests a deep-learning model for a remote PPG and respiratory-monitoring system using an off-the-shelf camera.

A remote PPG (rPPG) method can be used to monitor cardiac activity, which is a non-obtrusive measurement that can be obtained using a webcam without any additional sensor device [17]. Because the camera can capture the reflection of the light response to blood-flow changes from the skin, the rPPG signal can be obtained from a facial video [18,19]. Remote respiratory measurements can also be implemented by monitoring the movements of the head and chest in a video stream [20].

Research about the non-contact measurements of the heart and respiration rates has been conducted for several years [21,22,23]. The existing non-contact method mainly detects the skin-color changes in order to estimate HR, and for the prediction of RR, the head and chest movements are monitored, or it is extracted from the rPPG estimation. For these, the forehead and cheek areas are set as the regions of interest (ROIs) for the PPG estimation, and the nose or mouth are set for the prediction of respiration by investigating the warm or cool air flow through them [24]. A deep-learning study has also been conducted to extract the rPPG information from facial video streams. The Siamese rPPG network [25] was designed from the original Siamese network structure and comprises the same two convolutional neural networks (CNNs) with equal weights for two different inputs. The input is divided into two parts fed in the two networks; thus, even if one cannot be used, it can be replaced by another one, thus resulting in model stability and high performance. The input data to the proposed Siamese rPPG are from the forehead and cheek (with nose and mouse) areas of facial video streams, which could contain information on the heart and respiratory rates. 

Multitask learning is a learning paradigm that can improve the overall performance of multiple tasks, while simultaneously learning multiple related tasks in the same model. Compared with the conventional single-task learning model, this multitask learning process could take advantage of the previous learning experiences for the associated task, thus allowing us to update the model parameters. For biomedical applications, the multitask temporal shift convolutional attention network (MTTS-CAN) is a deep-learning model that implements on-device, contactless, vital measurements and simultaneously predicts the rPPG and respiratory signals [26]. The MTTS-CAN model is associated with complicated preprocessing. However, our proposed multitask Siamese (MTS) model has fewer parameters than the MTTS-CAN and is associated with simpler preprocessing.

We propose a multi-task Siamese (MTS) model, which can simultaneously estimate the remote PPG and respiratory signals using the same input image for the single-task Siamese network. The MTS model, which combines the advantages of the MTTS-CAN and Siamese network models to improve its performance during the execution of multiple tasks with high efficiency. Using the suggested model, rPPG and respiratory signals are simultaneously estimated with higher accuracy and fewer parameters compared with the conventional multitask learning model. The proposed MTS model is computationally lightweight due to the reduced the model parameters, which could be advantageous to implementing on-device learning and testing in a small edge device such as a mobile phone or a portable device [25,26,27]. This could realize the continuous and convenient HR- and RR-monitoring system in our daily lives and can be expected to improve the health conditions of patients with cardiac and respiratory disorders. 

Section 2 addresses the overall architecture of the MTS model with a detailed description, and Section 3 describes the experimental method used to evaluate the performance of the MTS model. In Section 4, the experimental results of the MTS model are compared with those of other remote PPG and respiratory models. In Section 5, we discuss why the performance of the MTS model is better than that of the single-task model (Siamese network with the convolutional block attention module (CBAM)). Finally, the conclusions are presented in Section 6.

## 2. Algorithm

Inspired by the Siamese rPPG network [25,28] and the MTTS-CAN [25], we propose the MTS, whose overall flowchart is illustrated in Figure 1. The structure of the MTTS-CAN could implement the multitask learning process to learn more than one task at the same time, that is, the simultaneous prediction of the PPG and respiratory signals. In the original MTTS-CAN architecture, the input image of the current and the previous frames should be provided together for its training, and the attention networks are also included, which is a complicated pretreatment process causing computational burden. On the contrary, Siamese networks require a simpler image-preprocessing process, for which the original Siamese rPPG network comprises a simple image-preprocessing step with low-computational complexity. In addition, the accuracy and stability of the rPPG prediction is improved owing to the weight sharing between the two branches of the Siamese networks. The MTS model is designed based on the structures of the Siamese rPPG network and the MTTS-CAN model and is expected to possess the advantages of both models. A major disadvantage of the conventional Siamese networks is the requirement of longer training time than the typical deep neural networks. Siamese networks are relatively slow to be trained, since two pairs of networks should be created during the learning process. Therefore, the implementation of the multitask learning model for the slow Siamese networks could be a solution for the long training issue due to simultaneously learning two tasks over a similar duration as the single-task model. In addition, it has a great advantage of reducing the number of model parameters by the simultaneous learning process of two tasks.

The MTS model in Figure 1 predicts cardiac (PPG) and respiratory signals using facial video streams, whereby two regions of interest (ROIs) of the forehead and cheek are separately selected as the inputs to the Siamese network. The cheek and forehead are the optimal areas used for capturing blood-flow changes and have often been used for rPPG predictions [27]. In particular, blood-volume pulses are usually obtained from the cheeks and forehead [25]. The proposed MTS model has multi-input and multi-output structures, whereby two different inputs from the cheeks and forehead separately enter the two branches of the Siamese networks. After the process of the CNN layers is completed, the outputs of the forehead and cheek streams are finally merged into one in the Add layer. The output of the Add layer yields a predicted PPG signal, and the respiratory signal can be obtained through the addition of a dense layer, as shown in Figure 2. The dense layer was not added after the PPG output because the performance was degraded by adding them. Additionally, a dropout layer was applied to reduce the probability of overfitting [29]. The activation function is the leaky rectified linear unit (Leaky ReLU) [30], formulated in Equation (1). Alpha is the slope of the leaky ReLU function, and α generally has a value of 0.01.

The ReLU function outputs 0 or 1 when it is less or greater than a predefined threshold value. This greatly reduces the multiplication, thereby reducing the probability of vanishing, exploding problems. Unlike the ReLU function, if all the differentiated values in any layer are 0, then learning does not proceed in any layer thereafter. The leaky ReLU has an output of the input value multiplied by 0.01 to prevent the knockout problem [30]. As shown in Figure 3, for learning negative data, the input value lower than the threshold is multiplied by 0.01.
(1)$$LeakyReLU\left(x\right)=\{\begin{array}{c} \;x,\;\;if\;x\;\geq 0\\ \;0.01x,\;if\;x<0\; \end{array}\;\;\;$$
where x ∈ [−∞,∞]. 

The CBAM [31] is an attention module used in image classification and object-recognition models. The CBAM has shown significant improvements in classification performance with few parameters. The CBAM effectively highlights and suppresses intermediate features using channel attention and a spatial attention module, respectively [31]. The CBAM has been applied to estimate PPG and respiratory signals from facial video streams, thus highlighting the significant features of the images, which increases the learning effect with few parameters.

We added a 1 × 1 convolution layer used in GoogLeNet for each convolution layer [32]. The 1 × 1 convolution layer adjusts the number of dimensions in order to design deep networks with fewer computations and time resources. The MTS model effectively reduces the parameters and makes a more lightweight model using the 1 × 1 convolution layer, making it available in various environments.

## 3. Methods

We compared the proposed MTS model with the MTTS-CAN, the original single-task Siamese rPPG Network [25], the Siamese network with CBAM, and 1 × 1 convolution layer. The results of the MTTS-CAN and Siamese networks with CBAM were produced using the COHFACE dataset, and those of the original Siamese rPPG network were obtained from [25], which also used the same COHFACE dataset.

### 3.1. Dataset

COHFACE is a remote photoplethysmography dataset. It is a dataset presented to evaluate the proposed algorithm in a standard and principled manner [33]. Compared to some other datasets, more realistic conditions are included. So, the experimental dataset is COHFACE, which includes 160 videos from 40 subjects recorded at 20 FPS with a resolution of 640 × 480 pixels. Each video was acquired for approximately 1 min using a Logitech HD C525 camera, and the subject’s face and upper body are shown. In addition, the PPG and respiration signals were synchronized with the video, and a Thought Technology device and BioGraph Infiniti software (Thought Technology, Ltd., Montreal, QC, Canada) were used for the analysis [25,34]. Among the four video streams obtained for each subject, light evenly entered the screen to clarify the facial images in two streams, while light was naturally exposed, and their faces were captured in a relatively dark environment in the other two streams [22]. The training, validation, and test datasets were divided at a ratio of 3:1:1 for the cross-validation of the model.

### 3.2. Pre/Postprocessing

To fix the input size of the model, the cheek and forehead images of the subjects in the dataset were extracted using the dlib’s face recognition library [35]. As shown in Figure 4, the forehead area was determined from the top of the head to the eyebrows and the cheek area from the tip of the nose to the chin. The areas were set in the first frame of the video and fixed throughout the entire 600 frames. Considering the ambient noise in the beginning frames of the video, 600 frames were extracted from the middle of the video streams. The true PPG and respiratory signals corresponding to each frame were obtained from the dataset.

After the estimation of the PPG and respiration signals using the deep-learning models, *HR* was calculated from the PPG signal using the fourth-order Butterworth bandpass filter, while *RR* was estimated from the respiration signal using the second-order Butterworth bandpass filter. The cutoff frequencies of the filters were (0.66 Hz, 3.3 Hz) for HR and (0.1 Hz, 0.4 Hz) for *RR*. *HR* (or *RR*) may be extracted from the distance between peak points. To derive the final *HR* and *RR*, the peaks of the filtered PPG and respiratory signals were detected, and the means of the intervals between the peaks were calculated using Equation (2) [5].
(2)$$HR\;or\;RR=\frac{60}{\sum_{i=2}^{N}\;\left(p_{i}-p_{i-1}\right)/N-1}$$
where *p* denotes the *i*th time instance of a peak in the PPG or respiratory signal, and *N* is the total number of peaks.

### 3.3. Environment and Evaluation

We used an NVIDIA GeForce RTX 2060 graphics processing unit to train the model, which was implemented in the TensorFlow framework [36]. The loss function used to train the model was the Pearson correlation coefficient expressed by Equation (3).
(3)$$r\left(x,\;y\right)=\frac{\sum_{i=1}^{N}\left(x_{i}-\underline{x}\right)\left(y_{i}^{\;}-\underline{y}\right)}{\sqrt{\sum_{i=1}^{N}{\left(x_{i}-\underline{x}\right)}^{2}\;\;\;\;}\sqrt{\sum_{i=1}^{N}{\left(y_{i}^{\;}-\underline{y^{\prime }}\right)}^{2}\;\;\;\;}}$$
where $x_{i}$ and $y_{i}$ are the ground truth and predicted values with size *N*, and $\underline{x}$ and $\underline{y}$ are their mean values. The Pearson correlation r(x, y) has a value between −1 and 1. The optimizer for the model training was Adam with a learning rate of 0.0001, β1 = 0.9, and β2 = 0.999 (β1, β2 are exponential decay rates for the moment estimates) [37]. The loss function in Equation (4) was applied to train the deep-learning models.
(4)Loss=1−r(x, y)

To evaluate the performance of the models, the metrics of the Pearson correlation coefficient (*R*), mean absolute error (*MAE*), and root-mean-square error (*RMSE*) were calculated, as described in Equations (5)–(7).
(5)$$R=\frac{\sum_{i=1}^{N}\left(X_{i}-\underline{X}\right)\left(X_{i}^{\prime }-\underline{X^{\prime }}\right)}{\sqrt{\sum_{i=1}^{N}{\left(X_{i}-\underline{X}\right)}^{2}\;\;\;\;}\sqrt{\sum_{i=1}^{N}{\left(X_{i}^{\prime }-\underline{X^{\prime }}\right)}^{2}\;\;\;\;}}$$
(6)$$MAE=\frac{\sum_{i=1}^{N}\left|\;X_{i}-X_{i}^{\prime }\;\right|}{N}$$
(7)$$RMSE=\frac{\sum_{i=1}^{N}{\left(X_{i}-X_{i}^{\prime }\right)}^{2}}{N}$$
where $X_{i}$ denotes the *HR* (or *RR*) estimated from the predicted PPG (or respiration) signal, and $X_{i}^{\prime }$ is the *HR* (*RR*) obtained from the ground-truth PPG (or respiration) signal. $\underline{X}$ and $\underline{X^{\prime }}$ are the averages of the predicted and real heart rate or respiratory rate (real *HR*s or *RR*s), respectively. We trained the MTS model within 250 epochs with a batch size of one and dropout rates of 0.25, 0.5, and 0.6. The single-task Siamese network was also trained under the same conditions as the MTS. The MTTS-CAN was trained for 30 epochs with a batch size of two. This is because the test results were the best when learning was implemented in this manner.

## 4. Results

To evaluate the performance of the proposed MTS model, its performance was compared with those of the existing Siamese rPPG network [25], the Siamese network with CBAM, and the MTTS-CAN [26]. The Siamese network with the CBAM includes the CBAM attention mechanism with the existing Siamese rPPG network and reduces the model parameters by adding a 1 × 1 convolution layer [32]. Inspired by MobileNet, which was successfully made to be lightweight in order to use deep learning on edge devices, with 1 × 1 Conv, it has an advantage in the computation over the standard convolutional layer, resulting in the lightening of the model [34]. These Siamese network models are single-task learning models for only one task. As a benchmark test model for multitask learning, the MTTS-CAN was compared, which was trained within 30 epochs using a batch size of two.

Table 1 lists the benchmark test results, including the proposed model, the Siamese network with the CBAM, and the other conventional single learning models. The Siamese network with the CBAM of Table 1 was used to extract the HR as a single-task model improved through CBAM and 1 × 1 conduction layer, etc., referring to the Siamese rPPG network [25]. Given that the Siamese network with the CBAM is a single-task model, learning the HR and RR was attempted. Unfortunately, the Siamese network-based models could not learn the respiratory signals owing to their original designs for the PPG signals; thus, their results could not be included in the table. We will talk about this later in discussion. Table 2 lists the performance measurement of each HR and RR by comparing the multitask model with our proposed model, the MTS. Table 3 lists the number of parameters of the benchmarking models. The MTS model could yield a comparable HR performance with fewer parameters compared with the single-task models. In particular, the R value of the MTS was higher than that of the original Siamese network, even though the number of parameters was 16 times lower. The Siamese network with the CBAM produced a performance similar to that of the MTS with a considerably reduced number of parameters compared with the original model. The number of parameters of the Siamese network with the CBAM was slightly smaller than that of the MTS, but this is because the Siamese network with the CBAM is a single-task model. In addition, the multitask learning model, the MTTS-CAN, was also outperformed by the MTS model for the predictions of HR and RR, which required smaller numbers of parameters.

The correlations between the predictions and their ground truths (real HR) obtained using the proposed MTS and the conventional multitask model MTTS-CAN are displayed as scatter plots in Figure 5. Figure 5a shows that the predicted HR using the MTS produced a high correlation coefficient with the real HR ($R^{2}$ = 0.94). Conversely, Figure 5b shows the correlation results of the HR for the MTTS-CAN model, whereby the $R^{2}$ value of the HR was 0.08. This means that the MTS describes HR data much better than the MTTS-CAN. These results demonstrate that the proposed MTS model overwhelmingly outperforms the conventional multitask learning model MTTS-CAN. Although the $R^{2}$ value of the respiratory rate was lower than that of the HR, the MTS model was able to significantly reduce the MAE and RMSE of the predicted respiratory signals, compared with those of the MTTS-CAN.

Figure 6 illustrates the predicted PPG and respiratory signals of the four subjects with their ground truths (PPG and respiratory signals). Although the predictions of the PPG signals appear more accurate than those of the respiratory signals, the similar peak patterns of their predictions are noticeable in all figures, thus resulting in reasonable predictions of HR and RR. The results in Figure 4 and Figure 5 demonstrate that the proposed MTS model could produce a similar performance to that of the original Siamese rPPG network with an extremely small number of parameters. It is also noted that the MTS model could simultaneously yield reasonable predictions of the RR and HR through its multitask learning architecture, while only the HR could be estimated using the original Siamese rPPG network. 

It has been demonstrated that the proposed MTS model outperforms the single-task Siamese networks with the CBAM. For a more rigorous investigation of the MTS and single-task Siamese network model’s learning processes, their learning curves were generated and are illustrated with their loss values in Figure 7. As can be seen in Figure 7c, the MTS has only one learning curve for the predictions of both PPG and respiratory signals owing to its multitask learning architecture. Similar to the results in Table 1, the single-task Siamese network model continuously produced an increasing validation loss as the epoch increased (see the blue line in Figure 7b). Because the single-task model is only designed for the prediction of the PPG signal, the networks may need to be completely redesigned to learn the respiratory signal. Conversely, the MTS was successful in simultaneously learning the respiratory signal as well as the PPG signal with a small number of parameters.

## 5. Discussion

Previous studies on the estimation of RR using video cameras mainly relied on the monitoring of the abdomen or chest movements [41,42]. This approach requires an additional camera for the application, where both HR and RR need to be simultaneously monitored using noncontact visual sensors. The proposed multi-task learning model could reduce the additional equipment, as well as the additional model that only estimates the RR since it deals with completely different areas of the body compared with those for the HR estimation.

The proposed MTS model was successfully trained using the facial videos together with the PPG and respiratory signals, yielding the accurate estimation of the HR and RR. Previous studies [11,12] have reported that PPG signals could contain respiratory information, and Nakajima et al. [43] also demonstrated the extraction of respiratory rates from PPG signals. Thus, the accurate estimation of rPPG using the proposed model from the facial videos could include the respiratory feature as well as the heartbeat information. Additionally, the MTS model was trained using the facial videos together with the respiratory signals recorded from the chest, in order to monitor the movement of facial parts such as the nostrils to extract the respiratory information. For future research, we will try to look into the relationship between the movement of a specific facial part and the respiration in order to check this.

The single-task Siamese rPPG network produced superior results of HR estimation in terms of MAE and RMSE compared with those using the proposed MTS model. Their performance difference could be understandable owing to it being designed only for HR estimation. However, the MAE and RMSE of the MTS would be comparable with meaningful levels of less than 5, which can be found in Table 4 showing the benchmark results of the various rPPG-prediction models using the COHFACE dataset. In addition, its correlation coefficient was higher than that of the Siamese rPPG network. Considering the number of parameters of the models shown in Table 3, the size of the MTS model was 16 times smaller than that of the single-task learning model, which is a significant advantage of using the proposed multitask learning model, especially for edge devices when conducting multiple tasks. In addition, the MTTS-CAN is also designed to reduce the model size; as can be seen in Table 3, it is 12 times smaller than the Siamese rPPG network. Despite this, the proposed MTS model contains a smaller number of parameters than that of the MTTS-CAN, and significantly improves the estimation performance of HR and RR, as demonstrated in Table 2.

The proposed model selects regions of interest (ROIs) in the facial image before the training process, for which the cheeks and forehead were chosen as the ROIs. Recently, a study compared the results using selected ROIs and the entire face areas for the rPPG model training, which reported that the model trained using the entire areas outperformed the other [45]. They demonstrated that skin-color changes in all facial areas would be helpful to yielding accurate rPPG signals. For future work, this model will be trained using the full facial images, and we will try to decrease the model size while dealing with the increased input data size. 

Through the Siamese network, heart rate and respiratory rate were successfully predicted. The heart-rate information has a significant effect on the prediction of respiratory rate, which helps improve accuracy [11,12]. However, there is a lack of evidence that information on respiratory rate helps predict heart rate.

We applied the ROI detection method from the facial area to train the model. The cheek and forehead areas were extracted as the areas of interest, which were used for the training dataset. A recent study has investigated the performance of the rPPG estimation corresponding to different areas of the face including the entire face [49]. It reported that the more areas of the face the model learns, the better the rPPG-prediction performance it could produce owing to the sufficient information on the skin-color changes and movements of the subjects. Therefore, further investigation is needed to confirm the optimal areas of the face for the rPPG prediction. It has been said that using as many areas as possible to learn skin-color change and movement is advantageous for learning. There is a need for further research on whether it is better to use the entire face or to extract and use a specific part [50].

The COHFACE dataset is an open dataset with a well-refined environment. The experiment was conducted with the subject’s movement or ambient noise as controlled as possible. However, what we want is to predict heart rates well in our daily lives. In future studies, it is necessary to analyze noisy data. Previous studies have shown that NIR cameras are strong in color change and movement [51,52]. In future research, we may consider combining the NIR camera with our MTS model.

## 6. Conclusions

We proposed a multitask Siamese network model that simultaneously estimates the PPG and respiratory signals from human facial video streams with only the use of a camera. We applied the Siamese network and the CBAM to the multitask learning architecture with a 1 × 1 convolution layer to considerably reduce the number of parameters in the model, while concurrently increasing its performance. It was demonstrated that the proposed model outperformed the single-task model as well as the conventional multitask learning model for RR estimation and exhibited comparable HR-prediction performance to that of the single-task model without increasing the number of model parameters. The MTS is a light model with a few parameters and could be suitable for on-device learning architectures.

## Figures and Tables

**Figure 1 sensors-22-05101-f001:**
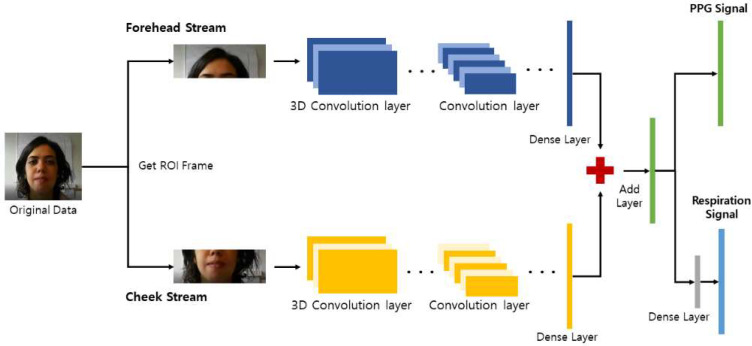
Multitask Siamese (MTS) network architecture. The forehead and cheeks are extracted as regions of interest (ROIs) from the facial image, which enter the weighted networks as forehead and cheek streams, respectively. Cardiac (photoplethysmography (PPG)) signals can be obtained based on the value from the last layer of the Siamese network, and the respiratory signal is estimated after an additional dense layer in the last part of the Siamese network.

**Figure 2 sensors-22-05101-f002:**
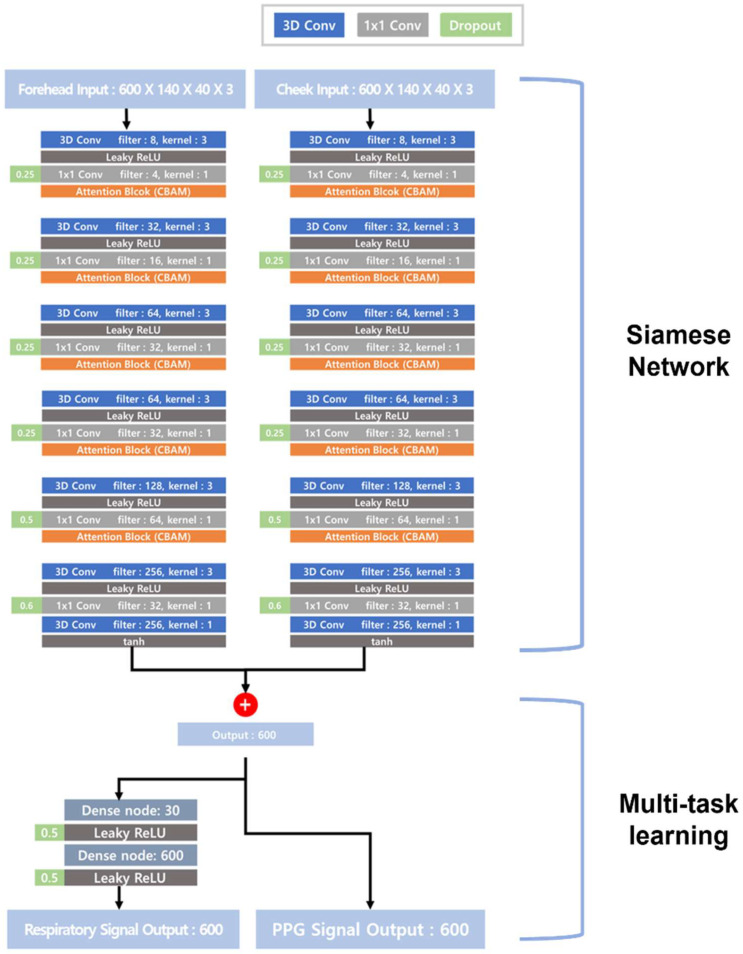
Structures of multitask Siamese network MTS. The input data are 140(w) × 40(h) images of the forehead and cheek as three RGB channels with 600 frames. The dimension of the input data is ‘frame × width × height × channel’. The output shows the PPG and the respiratory signals as 600 frames.

**Figure 3 sensors-22-05101-f003:**
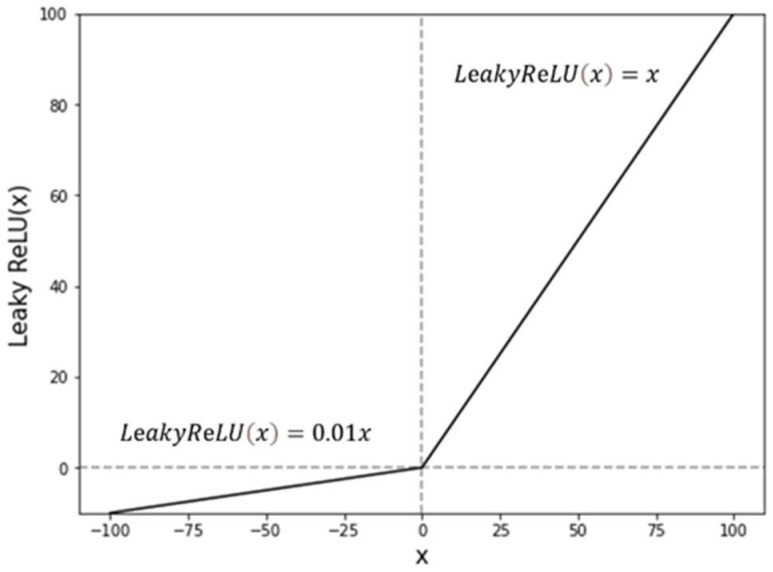
Leaky ReLU function.

**Figure 4 sensors-22-05101-f004:**
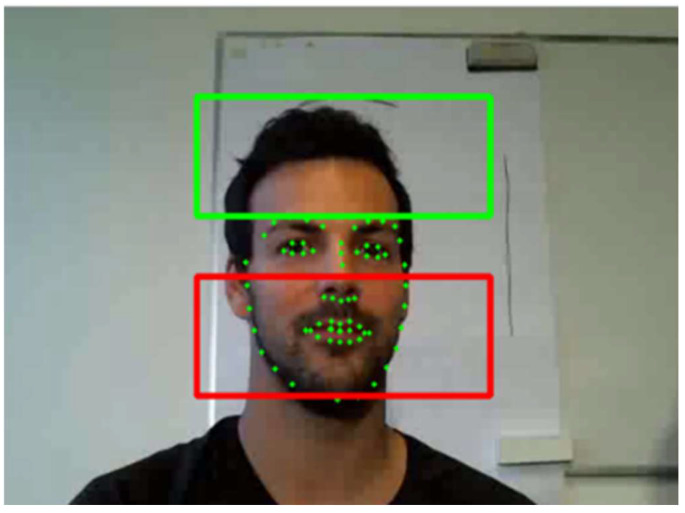
Preprocessing of the COHFACE dataset. Green dots are decided by the facial recognition library of “dlib”. The forehead and cheeks of the subject are detected by the green and red areas shown above.

**Figure 5 sensors-22-05101-f005:**
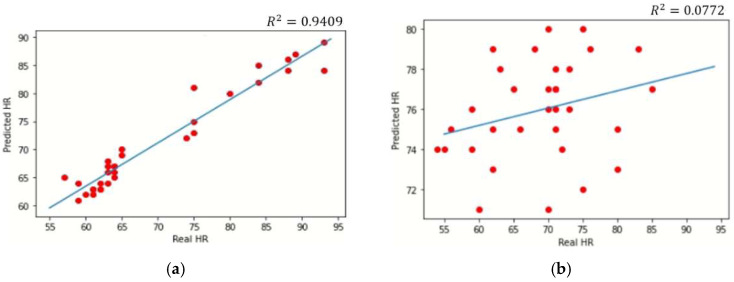
The HR correlation results. (**a**) HR correlation results predicted using the proposed MTS model. (**b**) HR correlation results predicted using the MTTS-CAN model. Scatter plots display the relationship between the HR predictions and their true labels for the proposed MTS model and the other conventional models. MTS produced relatively comparable or even higher correlations for the predictions of HR compared with the other models.

**Figure 6 sensors-22-05101-f006:**
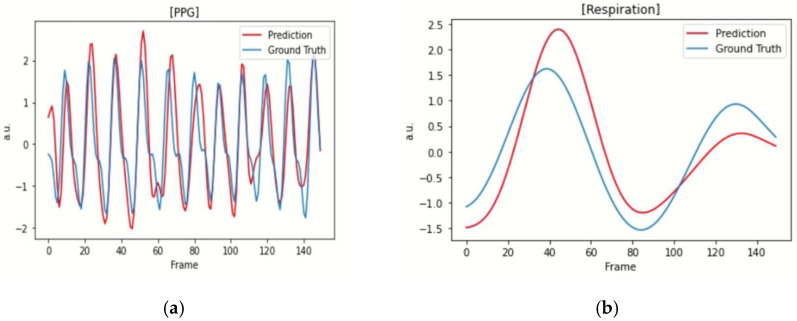
Examples of the predicted PPG and respiratory signals. (**a**) An example of the predicted PPG signal. (**b**) An example of the predicted respiratory signal. Predicted PPG and respiratory signals are plotted in red lines and their true labels in blue lines. Note the high similarity of the MTS model’s predictions to the true labels.

**Figure 7 sensors-22-05101-f007:**
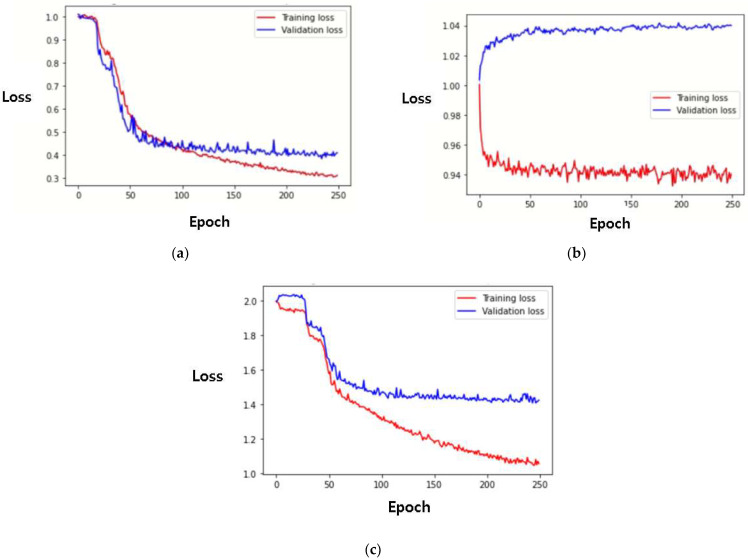
Learning curves of the rPPG-prediction models. (**a**) Learning curves of the Siamese network with convolution block attention module (CBAM) for the prediction of the PPG signal. (**b**) Learning curve of the Siamese network with CBAM for the prediction of the respiratory signal. (**c**) Learning curve of MTS for the predictions of both PPG and respiratory signals. Learning curves (shown as loss) of the Siamese network with CBAM and MTS for the predictions of PPG and respiratory signals are displayed over a span of 250 epochs.

**Table 1 sensors-22-05101-t001:** Benchmark test results comparing our single-task model with the latest research results within five years. Heart rates (HR) are calculated in beats per minute (BPM), and the model performances are evaluated using the metrics of Pearson correlation coefficient (R), mean average error (MAE), and root-mean-squared error (RMSE).

Model	HR (BPM)		
	R	MAE	RMSE
Siamese rPPG network [25]	0.73	0.70	1.29
Model by Z.-K. Wang et al. [38]	0.40	8.09	9.96
ETA-rPPGNet [39]	0.77	4.67	6.65
Model by Y.-Y. Tsou et al. [40]	0.72	0.68	1.65
**Siamese network with convolution block attention module (CBAM) (proposed)**	**0.97**	**2.31**	**3.29**

**Table 2 sensors-22-05101-t002:** Benchmark test results comparing multitask Siamese network model (MTS) with other multitask models. Heart rates (HR) and respiration rates (RR) are calculated in beats per minute (BPM) and respiration per minute (RPM), and the model performances are evaluated using the metrics of Pearson correlation coefficient (R), mean average error (MAE), and root-mean-squared error (RMSE).

Model	HR (BPM)			RR (RPM)	
	R	MAE	RMSE	MAE	RMSE
Multitask temporal shift convolutional attention network (MTTS-CAN) [26]	0.20	7.97	10.38	9.0	9.50
**Multitask Siamese network (MTS, proposed)**	**0.97**	**2.84**	**3.52**	**4.21**	**4.83**

**Table 3 sensors-22-05101-t003:** The number of parameters of the multitask Siamese network model (MTS) compared with the other models for the predictions of the heart rates and respiration rates.

Model	# of Parameters
Siamese rPPG network [25]	11.80 M
Multitask temporal shift convolutional attention network (MTTS-CAN) [26]	0.93 M
**Siamese network with convolution block attention module (CBAM) (proposed)**	**0.69 M**
**Multitask Siamese network (MTS, proposed)**	**0.72 M**

**Table 4 sensors-22-05101-t004:** Benchmark performance of the various rPPG-prediction models on the COHFACE dataset [25]. 2SR, CHROME, and LiCVPR are the traditional signal processing-based methods, while HR-CNN and Two stream are data-driven and machine learning-based algorithms.

Method	MAE	RMSE
2SR [44]	20.98	25.84
CHROME [45]	7.80	12.45
LiCVPR [46]	19.98	25.59
HR-CNN [47]	8.10	10.78
Two stream [48]	8.09	9.96

## Data Availability

The data is available at: http://publications.idiap.ch/index.php/publications/show/3688 (accessed on 8 April 2022).

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
