# Peer review of "Multitask Siamese Network for Remote Photoplethysmography and Respiration Estimation"

_sensors, 2022, doi:10.3390/s22145101_

Round 1

Reviewer 1 Report

The manuscript “Multitask Siamese Network for Remote Photoplethysmography and Respiration Estimation” by Kwon et al. is dedicated to designing a neural network capable of simultaneous estimation of heart rate and respiratory rate using video photoplethysmography data.

While the article can be of interest to some audiences, the manuscript does not provide enough details about the experimental design to assess its merits. Thus, it is not acceptable for publication in the Sensors in the current form. My assessment is based on the following observations:

  1. In Fig 2, the forehead input is annotated as 600x140x40x3. The curious reader may imply that 600 is the number of frames from the text. However, other factors are not mentioned anywhere (width of the segment, the height of the segment, and the number of color channels?)
  2. The ground truth was mentioned several times (e.g., lime 264); however, no details were provided.
  3. Fig 5 does not make any sense for anyone who has seen a PPG signal. It looks like an over-filtered signal.

Minor comments:

  1. Multiple references are unacceptable in the current form. E.g., in [1] “available online” link does not make sense for non-Proguest users. In general, “available online” link should be complimentary to DOI, not just another link to the abstract. [3] does not name the source (WHO?), [28] does not provide any details.

Author Response

Original Manuscript ID: sensors-1697952

Original Article Title: “Multitask Siamese Network for Remote Photoplethysmography and Respiration Estimation”

To: MDPI sensors Editor

Re: Response to reviewers

Dear Editor,

Thank you for allowing a resubmission of our manuscript, with an opportunity to address the reviewers’ comments.

We are uploading (a) our point-by-point response to the comments (below) (response to reviewers), (b) an updated manuscript with yellow highlighting indicating changes (Supplementary Material for Review).

Best regards,

< HEEJIN LEE, JEONGHWAN LEE,  YUJIN KWON, JIYOON KWON, SUNGMIN PARK, RAYNGHEE SOHN AND CHEOLSOO PARK >

Reviewer 2 Report

1、In line 93,you have mentioned using fewer data compared with a conventional single-task learning model. However, the data used in this paper are all from an open-source dataset. Please give specific examples to illustrate the argument.

2、We would like you to expound the shortcomings of the original network to support your improved theory

3、We suggest that you could use some figures to illustrate the multi-tasking learning framework.

4、We noticed that you have added a 1 X 1 convolution layer to adjust the number of dimensions. The 1 X 1 convolution layer may increase the memory accesses in engineering. If implement the vital sign monitoring model in an edge device, whether the results become worse?

5、We'd like you to use specific experimental data in your conclusion to support your algorithm.

6、We encourage authors to add more evaluation indexes in ablation study, such as mean average error, instead of just loss curves.

7、There are obvious grammatical mistakes in Line 323.

Author Response

(The authors gave the same response as above.)

Reviewer 3 Report

"Multitask Siamese Network for Remote Photoplethysmography and Respiration Estimation" is an interesting article. The aim was to propose a multi-task Siamese (MTS) model, which could estimate the remote photo-plethysmography (rPPG) and respiratory signals simultaneously using the same input image for the single-task Siamese network. The results shown that the proposed methodology can simultaneously predict the heart and respiratory signals with the MTS model, while the number of parameters was reduced by 16 times with the mean average errors of heart and respiration rates, 2.84 and 4.21.

There are some issues in the article that need to be addressed.

Methods

Line 169. It is suggested to put the meaning of COHFACE.

Results

The results on the RR have a very low R2, and it can be seen in Figure 4d that the RR varies between 12.5 and 22.5 but the predicted RR only varies between 14 and 20. Are these results clinically useful? It is suggested to clarify these results and analyze them more deeply in the discussion.

Would it be possible to calculate the degree of precision and accuracy of the proposed methodology by comparing the actual results with the predicted ones?. It is suggested to use statistical tests that compare the degree of consistency between two diagnostic techniques such as the Kappa Index, sensitivity, specificity, positive predictive value, negative predictive value, etc.   

Discussion

This section requires more work; it is the weakest in the manuscript. Most of it shows learning curve results, so it really should be part of the results and not part of the discussion. Only in the last paragraph of this section is a superficial analysis of the results made and a single reference is mentioned. It is suggested to completely rewrite this section and carry out a deep and detailed analysis of each one of the results contrasted with the results obtained in previous studies with their corresponding references.

References

A careful review of each of the references is suggested. The names of the authors are written in capital letters (reference 6). Many references are missing page numbers. In many references, after the name of the magazine, the name of the publisher of said magazine is placed (generally not), etc.

Author Response

Dear Editor,

Thank you for allowing a resubmission of our manuscript, with an opportunity to address the reviewers’ comments.

We are uploading (a) our point-by-point response to the comments (below) (response to reviewers), (b) an updated manuscript with yellow highlighting indicating changes (Supplementary Material for Review). 

Best regards,

< HEEJIN LEE, JEONGHWAN LEE,  YUJIN KWON, JIYOON KWON, SUNGMIN PARK, RAYNGHEE SOHN AND CHEOLSOO PARK >

Round 2

Reviewer 1 Report

The manuscript “Multitask Siamese Network for Remote Photoplethysmography and Respiration Estimation” by Lee et al. has been improved during the revision, and the authors partially addressed my concerns. However, it is still not ready for publication.

1.      The authors mentioned “ground truth -> Real PPG.” However, I don’t understand this statement. How ground truth (or “real” in authors’ notation, see Fig 4) HR and RR were collected or derived? Were they measured by ECG and provided in the COHFACE dataset or derived from a video stream by some method? A brief description of the dataset is required (the number of samples, etc.). How many videos were used for training? For testing?

2.      Overall, the logic and flow are absent, particularly in the Introduction and algorithm sections. The authors jump between topics and insert pieces that are not logically linked. Some examples:

a.      Page 2, 2nd paragraph: “Among the heart signal monitoring approaches, photo-plethysmography (PPG) is a method used to detect the amount of blood flow using light sources, based on the detection of the amount of blood volume through peripheral blood vessel [9]. HR denotes the number of heart beats per minute estimated using the PPG signal. Respiratory signals were recorded during the inhalation and exhalation respiratory phases, and the respiratory rate was calculated using the period of the respiratory signal.”

b.      In the Algorithm section, the authors introduce MTS, then jump to MTTS-CAN, then to Siamese networks. Then to MTTS-CAN, then to MTS, then disadvantages of Siamese networks again. And all of that is in a single paragraph.

3.      The discussion section is still very weak.

4.      Most figures (Fig 4-6) are not in the format acceptable for publication. Panels should be labeled with letters a, b, c, etc. The titles of each panel should be described in the figure caption

5.      I don’t understand what Eq.1 means.

6.      The authors use HR and PPG interchangingly, which is confusing. E.g., in the Discussion, “The MTS model could be successful in learning both PPG and respiratory signals.”

7.       Discussion “However, the MAE and RMSE of MTS would be comparable with the meaningful levels less than 5…” what is that supposed to mean?

8. On page 2, 2nd paragraph, the authors wrote, “This respiratory rate could also be estimated using the PPG signal, as reported in previous studies [10, 11].” However, in the Discussion, they wrote, “Previous studies for the estimation of RR using video cameras mainly relied on the monitoring of the abdomen or chest movements [38, 39].”

Author Response

Dear reviewer,

Thank you for allowing a resubmission of our manuscript, with an opportunity to address the reviewers’ comments.

We are uploading (a) our point-by-point response to the comments (below) (response to reviewers), (b) an updated manuscript with yellow highlighting indicating changes (Supplementary Material for Review). 

Best regards,

Reviewer 3 Report

The new version of the paper "Multitask Siamese Network for Remote Photoplethysmography and Respiration Estimation" has some of the suggestions and adequate argumentation for the unrealized ones. The changes made have improved the quality of paper. 

Author Response

Dear, Reviewer

Thank you for helping to revise our manuscript.

Best regards,

Round 3

Reviewer 1 Report

The manuscript has been improved during the latest revision. However, it is not publication-ready yet.

Deficiencies:

1.      It is still not clear to me what are the respiratory signals in the COHFACE  dataset. It is stated that “Repiratory signals were recorded during the inhalation and exhalation respiratory phases,” which does not bring any value or clarity. Was “respiratory signal” extracted from the video?

2.      Eq. 1 still does not make sense to me. (alpha x, x) has multiple connotations in math in physics, but not what was described by the authors. Please revise it

3.      Figures 4 & 5: The titles of each panel are now present in the caption. Now, they need to be removed from panels. Leave only letters (a, b,..) to numerate the panels

4.      Page 6: technologies mentioned in “Thought Technology device and BioGraph Infiniti software” need to be referenced appropriately: company, country, etc

5.      The method of obtaining “Real HR” and “Real RR” needs to be spelled out explicitly.

6.      I can’t entirely agree with the authors’ response to my comment about using “HR” and “PPG” interchangingly. PPG is a signal collected using a photosensor, e.g., a camera. Multiple derived metrics (including HR) can be derived from it. Thus, the PPG signal has a broader meaning, and  PPG and HR cannot be used interchangingly. Please make changes to the text accordingly.

7.      Page 11: I don’t understand. “The MTS model could be successful in learning both PPG and respiratory signals, resulting in the accurate estimation of the HR and RR, considering the previous studies demonstrating the respiratory information in a PPG signal.” Please rephrase

8.    Multiple misspellings and inaccuracies:

a. Page 2: “the lung condition affected by infected diseases such as the flu and COVID19 [REF].” – missing reference

b. Page 2: The passage “Repiratory signals were recorded during the inhalation and exhalation respiratory phases, and the respiratory rate was calculated using the period of the respiratory signal. This respiratory rate could also be estimated using the PPG signal, as reported in previous studies [10, 11]” appears twice. Also, note the spelling of “respiratory.”

Page 2, 12: Abbreviations like “ppg” and “roi” need to be in capital letters

Page 3: “I was inspired to learn more than one TASK at the same time.” Probably the authors meant “It.” However, the whole sentence does not make much sense to me. I recommend combining it with the previous sentence.

Page 4: “pretreatment process.” Have you meant “pre-processing”?

Page 6: “So, The experimental” – uncapitalize “The”

Page  12 “However, the MAE and RMSE of MTS would be comparable with the meaningful levels less than 5, and its correlation coefficient is higher than that of Siamese-rPPG network. As shown in Table 4, various rPPG prediction models were benchmarked using the COHFACE dataset. It indicates that MAE and RMSE of 5 or less are good results.” – some repetition is present.

Page 12: The passage “We used the roi detection of the facial area for learning. We extracted the cheeks and forehead as areas of interest. And this area was used as learning data. In recent studies, there are results of comparing the results of learning by extracting the landmark of the face with the results of using the entire face for learning. Then, there is a study of the results that using the entire area of the face resulted in better results [42]. It was said that using as many areas as possible to learn skin color change and movement is advantageous for learning. There is a need for further research on whether it is good to use the entire face or to extract and use a specific part.” is present twice.

Author Response

Dear Reviewer,

Thank you for allowing a resubmission of our manuscript, with an opportunity to address the reviewers’ comments.

We are uploading (a) our point-by-point response to the comments (below) (response to reviewers), (b) an updated manuscript with yellow highlighting indicating changes (Supplementary Material for Review). 

Best regards,

< HEEJIN LEE, JEONGHWAN LEE,  YUJIN KWON, JIYOON KWON, SUNGMIN PARK, RAYNGHEE SOHN AND CHEOLSOO PARK >
